# Maternal Immunization and Antenatal Care Situation Analysis (MIACSA) study protocol: a multiregional, cross-sectional analysis of maternal immunization delivery strategies to reduce maternal and neonatal morbidity and mortality

Nathalie Roos,[1] Philipp Lambach,[2] Carsten Mantel,[3,4] Elizabeth Mason,[5] Flor M Muñoz,[6] Michelle Giles,[7] Allisyn Moran,[1] Joachim Hombach,[2] Theresa Diaz,[1] MIACSA expert advisory panel group

For numbered affiliations see end of article.

**Correspondence to**
Dr Nathalie Roos;
roosn@who.int

## ABSTRACT

**Introduction** Maternal immunization (MI) with tetanus toxoid containing vaccine, is a safe and cost-effective way of preventing neonatal tetanus. Given the prospect of introducing new maternal vaccines in the near future, it is essential to identify and understand current policies, practices and unmet needs for introducing and/or scaling up MI in low-income and middle-income countries (LMICs).

**Methods and analysis** The Maternal Immunization and Antenatal Care Situation Analysis (MIACSA) is a mixed methods, cross-sectional study that will collect data in four phases: (1) a review of global databases for selected health indicators in 136 LMICs; (2) a structured online survey directed at Maternal, Newborn and Child Health and Expanded Programme on Immunization focal points in all 136 LMICs; (3) semistructured telephone interviews of 30 selected LMICs and (4) 10 week-long country visits, including key informant interviews, health facility visits and focus group discussions. The principal analyses will assess correlations between the various aspects of MI delivery strategies and proxy measures of health systems performance related to vaccine-preventable disease control. The primary outcome will be a typology of existing MI delivery models, and secondary outcomes will include country profiles of child and maternal health indicators, and a MI gaps and needs analysis.

**Ethics and dissemination** The protocol was approved by the WHO Ethics Review Committee (ERC.0002908). The results will be made available in a project report and submitted for publication in peer-reviewed journals that will be shared broadly among global health decision-makers, researchers, product developers and country-level stakeholders.

## INTRODUCTION

Vaccine-preventable diseases are a major cause of global child morbidity and mortality,

### Strengths and limitations of this study

► The Maternal Immunization and Antenatal Care Situation Analysis (MIACSA) study provides a first time, comprehensive global overview and analysis of existing maternal immunization (MI) delivery strategies in low-income and middle-income countries (LMICs).

► The study benefits from a mixed-methods design; a multidisciplinary approach leveraging policy-level, academic and implementers' experience.

► Limitations include the small number of countries and healthcare facilities visited within each country included in the study, precluding generalisation of country visit findings to a national level.

► End-users' perspective is captured only indirectly through community health workers. Data on MI service delivery collected through an online survey targeting all LMICs are analysed within the limitations of the validity of data collected.

particularly in low-income and middle-income countries (LMICs).[1] Since the 1990s, public health interventions have more than halved under-five childhood mortality; however, reduction of stillbirths and of neonatal mortality (death in the first 28 days of life) has been slower.[2] This is in part due to the fact that most vaccines cannot be administered to newborns, who, being unable to develop protective responses due to limitations in their immune system, are left particularly vulnerable to infectious diseases. Vaccination of pregnant women, or maternal immunization (MI) with tetanus toxoid containing vaccine, has proven

to be an effective strategy to reduce neonatal tetanus, and is a potential strategy to reduce the burden of other vaccine-preventable diseases in mothers and infants. Thus, MI is one of the several strategies that aim to reach the third sustainable development goal of ending preventable maternal and newborn deaths.[3–5]

Studies have shown that MI can effectively protect the mother, as well as her child, through transplacental transfer of maternal IgG to the fetus.[6 7] The Maternal and Neonatal Tetanus Elimination (MNTE) initiative has led the way in the implementation of MI, combining at least two doses of tetanus toxoid-containing vaccine (TTCV) during pregnancy (TT2+) with the promotion of hygienic delivery and clean cord care practices, as well as vaccination of children and women of reproductive age, to eliminate maternal and neonatal tetanus (MNT) as a public health problem. Between the late 1980s and 2015, the MNTE initiative reduced global tetanus-related neonatal mortality by 96%.[8]

TTCV and inactivated influenza vaccines are considered safe and effective for use during pregnancy,[9] and are recommended for pregnant women by WHO.[6 10–13] New vaccines, several of which are under development and evaluation, target other important pathogens, such as group B streptococcus and respiratory syncytial virus, and may provide safe and cost-effective protection of mothers and their infants through MI in the future.[14–18]

In order to identify the challenges of implementing current and new vaccines for MI, a better understanding is needed of the capabilities and limitations of existing delivery platforms, such as antenatal care (ANC) services and the Expanded Programme on Immunization (EPI).[19] The capacity of ANC services to deliver vaccines to pregnant women will require thorough assessment, as globally, only 62% of women benefit from at least four ANC visits, that is, the proportion of pregnant women who received four or more ANC visits during their last pregnancy (ANC4+), and in Sub-Saharan Africa and South Asia, ANC4+ coverage is only 52% and 46%, respectively.[20] Delivering vaccinations and other essential interventions to women at the necessary timely intervals during pregnancy, as well as documenting the coverage and outcomes of such interventions, requires a robust ANC platform with sufficient personnel and resources.[21]

WHO recommends that pregnant women living in high-risk areas are sufficiently immunised against tetanus in order to protect the women and their newborn infants. MI with TTCV is routine in many countries[22 23]; however, the progress of tetanus vaccination in LMICs has faced challenges leading to delays in elimination, and uptake among pregnant women of other vaccines, such as influenza and pertussis vaccines, has been low. As a part of EPI services, routine tetanus immunization during pregnancy has been complemented with supplementary immunization activities in a majority of countries in order to reach high coverage and achieve MNTE goals. A better understanding of MI in the context of both ANC and EPI, including implementation of guidelines and policies, ministerial responsibilities at national and subnational levels, vaccine management including cold chain and logistics, vaccine administration, staff capacity, social mobilisation, vaccine acceptance and assessment of vaccine safety, may help to identify service delivery challenges as well as opportunities to optimise current and future MI efforts.[24]

Closer collaboration between ANC and EPI services could provide a unique and cost-effective opportunity to further strengthen preventive healthcare measures for women and children under each programme, by reducing missed opportunities for vaccination, including MI, as well as reinforcing the delivery of essential healthcare services.

In view of recent product and policy developments, WHO, supported by the Bill & Melinda Gates Foundation, aims to identify the knowledge gaps in MI delivery strategies by mapping the strengths and challenges of existing ANC and EPI services for pregnant women in LMICs through the Maternal Immunization and Antenatal Care Situation Analysis (MIACSA) project. The results will provide the evidence for a typology of MI delivery models, as well as identify the capacity needs and key system changes required to introduce new maternal vaccines and/or strengthen vaccine delivery for MI in LMICs. Ultimately, the project aims to identify and understand current MI-related and ANC-related policies, practices and the need for strengthening maternal child healthcare services, and how they could accommodate new MI vaccines.

## METHODS AND ANALYSIS
### Patient and public involvement
The development of the research questions was influenced by an interdisciplinary group of international experts for the MIACSA project. The project did not include patients, but restricted itself to national level programme managers and health facilities where health workers responded to interviews in their professional capacity.

### Study design and data collection
Between November 2016 and December 2018, a mixed-methods, cross-sectional study will be carried out in four phases to assess key health system features related to the implementation of MI (figure 1). An expert advisory panel (EAP), consisting of specialists in immunisation, maternal and neonatal health, MI implementation and social sciences, will provide technical advice on the study design, the development of research questions and surveys, the data collection methods and the results interpretation. In addition to following WHO standards for global monitoring surveys, all data collection tools and standard operating procedures will be reviewed and endorsed by the EAP. The surveys and country visits will be conducted in local languages when needed.

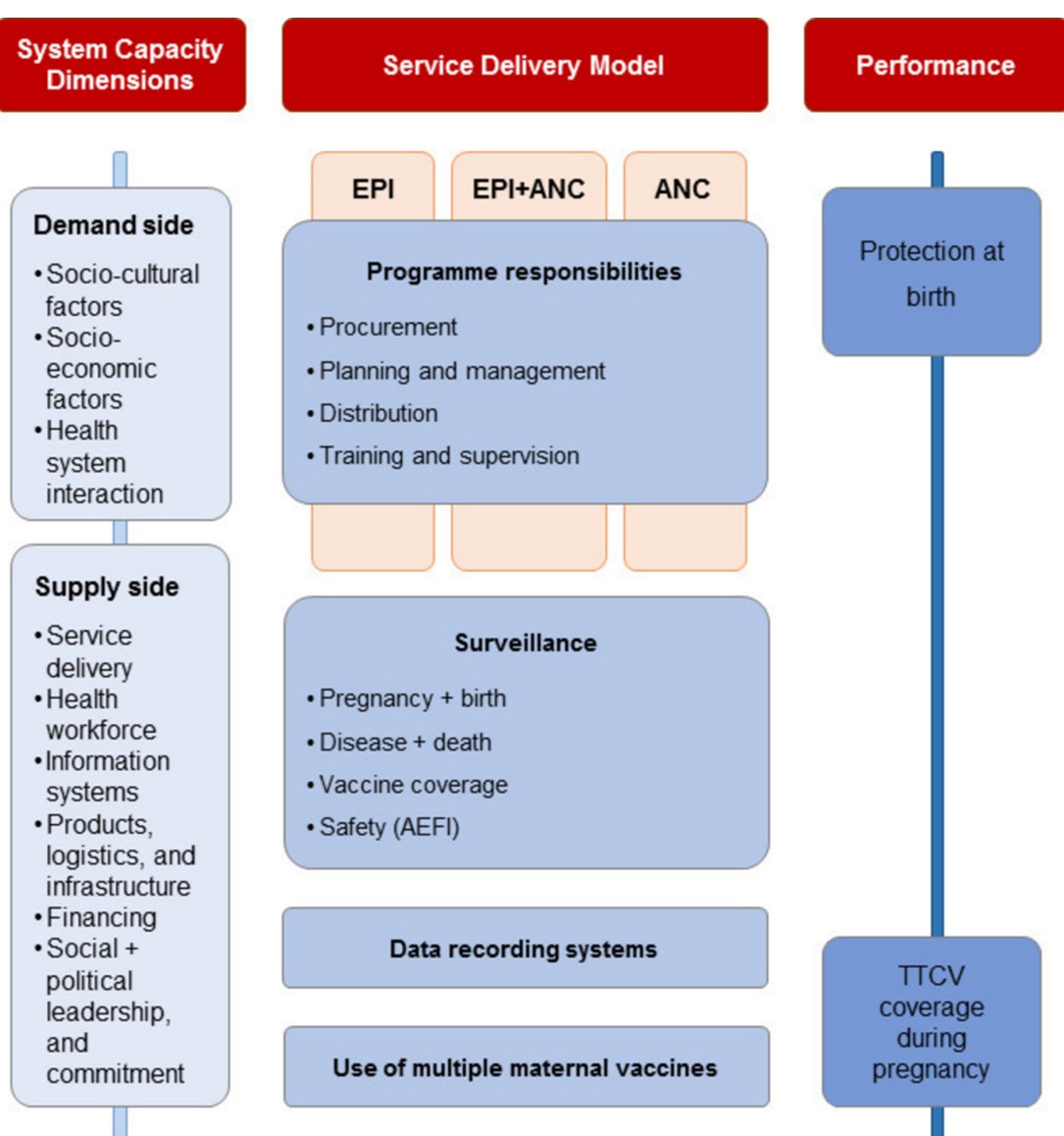

**Figure 1** Key health system features studied by the MIACSA project. AEFI, adverse events following immunisation; ANC, antenatal care; EPI, Expanded Programme on Immunization; MIACSA, Maternal Immunization and Antenatal Care Situation Analysis; TTCV, tetanus toxoid.

### Desk review of global data (data collection phase 1)

The first phase will consist of collecting key health indicators of LMICs to create outlines of country profiles, focusing on ANC and EPI services. A desk review of predefined health indicators (figure 2) from 136 LMICs will be conducted from existing global data sources, including Demographic and Health Surveys (DHSs)/ Multiple Indicator Cluster Surveys (MICSs), WHO/ United Nations Children's Fund (Unicef) estimates of national immunization coverage, WHO/Unicef Joint Reporting Forms (JRFs), MNTE reports and WHO Maternal, Newborn, Child and Adolescent Health (MNCAH) policy survey. The indicators will focus on governance and policy environment, health systems performance and immunisation activities, including MI. Data from phase 1 will be compiled in a database for analysis of the study's research questions, and will inform the selection of countries for phase 3 (see figure 2).

### Global online survey (data collection phase 2)

The country profiles established in phase 1 will be completed and, if needed, updated by an online survey with WHO Regional Offices, national Ministry of Health (MoH) focal points from Maternal, Newborn and Child Health (MNCH) and EPI programmes and their WHO Country Office counterparts in all LMICs, using a structured questionnaire (figure 3). Data will be collected on service delivery models of maternal tetanus vaccination, including delivery platforms, programme funding, disease surveillance and vaccine safety surveillance. Data on maternal vaccines other than tetanus will be included when pertinent. A draft questionnaire will be piloted in advance of the survey. Non-responders will be followed-up by telephone and email. Revisions following queries on missing, erroneous or inconsistent data will be done at country level.

**General**
- World Bank income classification;
- Female literacy rate.

**Health systems**

General
- Health systems classification;
- Birth cohort (most recent year of available data);
- Target population of pregnant women.

Governance and policy environment
- Percentage of total expenditure on routine immunization financed by government funds;
- Existence of national immunization technical advisory group (NITAG);
- National policy on minimum antenatal care (ANC) visits;
- Eligibility for global vaccine alliance (GAVI) support.

Health systems performance
- Maternal, neonatal, and infant mortality;
- Stillbirth rates;
- Physician and midwife densities;
- Institutional deliveries;
- Coverage of a minimum of four ANC visits (ANC4+).

Immunization, including maternal immunization
- Number of confirmed tetanus and neonatal cases;
- Coverage of at least 2 doses of TTCV during pregnancy (TT2+);
- TTCV as a proportion of CES and WUENIC vaccines, i.e. BCG, DPT1, DPT3, HepB1, HepB3, Hib1, Hib3, MCV1, MCV2, PAB, PCV1, PCV3, Pol1, Pol3, TT1, TT1+, TT2, TT2+, TT3 TT4, TT5, RCV1, RotaC, YFV;
- TTCV administered to pregnant women during routine visits;
- Most recent TTCV supplementary immunization activities (SIA), age range and size of target population, vaccination coverage, vaccine presentation, and year of next planned activity;
- Number of adverse events following immunization (AEFI);
- Maternal and Neonatal Tetanus Elimination (MNTE) status (year of elimination);
- Influenza vaccine administered to pregnant women;
- Pertussis vaccine administered to pregnant women.

Immunization-associated activities
- Vitamin A supplementation.

**Figure 2** Study phase 1: list of indicators for the review of global databases. CES, coverage evaluation survey; DPT1, first dose of diphtheria–pertussis–tetanus vaccine; DPT3, third dose of diphtheria–pertussis–tetanus vaccine; HepB1, first dose of hepatitis B vaccine; HepB3, third dose of hepatitis B vaccine; Hib1, first dose of *Haemophilus influenzae* type B vaccine; Hib3, third dose of *H. influenzae* type B vaccine; MCV1, first dose of measles-containing vaccine; MCV2, second dose of measles-containing vaccine; PCV1, first dose of pneumococcal conjugate vaccine; PcV3, third dose of pneumococcal conjugate vaccine; Pol1, first dose of polio-containing vaccine; Pol3, third dose of polio-containing vaccine; RCV1, first dose of rubella-containing vaccine; RotaC, second or third dose of rotavirus vaccine depending on number of doses recommended in national schedule; TT1, first dose of tetanus toxoid vaccine; TT1+, at least one dose of tetanus toxoid vaccine; TT2, second dose of tetanus toxoid vaccine; TT2+, at least two doses of tetanus toxoid vaccine; TT3, third dose of tetanus toxoid vaccine; TT4, fourth dose of tetanus toxoid vaccine; TT5, fifth dose of tetanus toxoid vaccine; WUENIC, WHO/United Nations Children's Fund (Unicef) estimates of national immunization coverage; YFV, yellow fever vaccine.

### Telephone interviews (data collection phase 3)

In order to understand how existing healthcare delivery services could be adapted to implement MI beyond tetanus immunization, further data will be collected on delivery platforms for maternal tetanus vaccination in LMICs. In-depth telephone interviews will be conducted with EPI and MNCH programme officers responsible for MI at the MoH in a sample of 30 countries, using

**Service delivery models**

Routine maternal tetanus vaccination

- Policy content and coverage data;
- Existing delivery models, e.g. facility-based ANC and EPI/immunization services, outreach services, and regular and ad hoc health campaigns;
- Type(s) of TTCVs administered, i.e. TT, Td, Tdap (adult formulation).

Integrated health campaigns for maternal tetanus vaccination

- Programme management and coverage data;
- Existing campaigns integrated with vaccination, e.g. deworming, vitamin A, malaria, nutrition;
- Past and future schedules of integrated health campaigns.

EPI, ANC or other organisation of maternal tetanus vaccination

- National level coordination planning and management;
- Training (rationale, safety and AEFI surveillance) and supervision of vaccinators;
- Vaccine procurement and distribution;
- Monitoring and evaluation, i.e. records (ANC or EPI-based personal, clinic, or electronic), frequency of performance assessment, monitoring indicators, e.g. TT2+, PAB.

Funding for maternal tetanus vaccination programme

- Domestic and external funding.

Disease surveillance

- Maternal and neonatal tetanus, i.e. passive, sentinel, active, community-based;
- Other health indicators, i.e. congenital rubella syndrome, neonatal sepsis, neonatal mortality, maternal mortality, BCG at birth, OPV at birth, HBV at birth, other.

ANC capacity for maternal immunization

- Policy for ANC, i.e. number of visits, settings for ANC provision, i.e. government or private health facility/hospital, clinic, outreach programme;
- Any user fees for ANC and maternal immunization.

**Vaccine safety surveillance**

- Surveillance of AEFI in general and maternal immunization;
- Any available surveillance data.

**Other maternal vaccines**

- Routine maternal immunization, e.g. influenza, pertussis, or other;
- Programme management, i.e. EPI, ANC, or other responsible for planning, training, supervision, procurement, and distribution.

**Figure 3** Study phase 2: variables collected from online survey of 136 low-income and middle-income countries (LMICs). A structured questionnaire will be used to determine which service delivery platforms are in place for tetanus vaccination of pregnant women in LMICs, and to understand how existing health services could be adapted to implement maternal immunisation beyond tetanus vaccination. Internal validation questions are incorporated in the questionnaire, and sources of data are requested, that is, if administrative data or personal estimates. AEFI, adverse events following immuni; ANC, antenatal care; EPI, Expanded Programme on Immunization; HBV, hepatitis B vaccine; OPV, oral polio vaccine; PAB, protection at birth; Td, tetanus diphtheria; Tdap, tetanus diphtheria–acellular–pertussis; TT, tetanus toxoid; TT2+, at least two doses of tetanus toxoid vaccine during pregnancy.

a semistructured questionnaire (figure 4). The countries will be selected based on the performance of MI as assessed by coverage of maternal TTCV and ANC, geographic representation and recommendations from WHO Regional Offices on MI priorities. The countries will be stratified into four groups; high and low maternal tetanus vaccination performance measured as protection at birth (PAB), that is, the proportion of newborns

**Country context**

Overview

- Integration of ANC and EPI organisation, i.e. national level coordination of maternal immunization, representation of maternal and newborn health care experts in NITAG;
- National policy and action plan for maternal immunization, and respective targets, i.e. coverage, completeness and timeliness of reports, how and why targets are/are not met;
- Existence of national HMIS, completeness and mode of data collection, available data.

Funding of maternal tetanus immunization and ANC services

- Domestic and external funding of ANC services and maternal immunization, user fees for ANC and tetanus immunization, and impact of funding situation on ANC and/or maternal immunization, e.g. procurement, logistics, training, mobilisation, and/or administration.

Human resources

- National and district level coordination and challenges for delivery.

**Service delivery through ANC and the birth context**

- Coverage and quality of ANC, i.e. staffing, coverage, precision of estimate of gestational age, counselling, prevention and interventions, referral systems, and outreach services;
- Challenges to ANC delivery, e.g. staffing, equipment, infrastructure;
- Information used for planning and prioritisation, e.g. coverage, staffing, funding, user needs;
- ANC records, i.e. verbal, written, electronic, and personal or facility-based, follow-up.

**Tetanus vaccine delivery to pregnant women**

Overview

- Type of vaccines delivered, i.e. TT, Td or other, frequency, any integration with ANC/EPI;
- Private providers of tetanus vaccination of pregnant women, available data;
- Existing quality of TT vaccine cold chain, and ANC services' capacity for vaccine storage;
- Current vaccination of pregnant women through ANC, staffing and challenges, e.g. infrastructure, cold chain, vaccine supply, skilled staff.

Vaccination of pregnant women outside ANC

- Primary, secondary and tertiary clinical settings for vaccination of pregnant women, staffing;
- Information used for planning and prioritisation of outreach services, e.g. ANC coverage, staffing, funding, user needs.

Recording of tetanus immunization during pregnancy

- Policy, guidelines, operating procedures, with attention to immunization history and dosage.

Maternal and neonatal tetanus surveillance

- Existing neonatal and maternal tetanus surveillance systems, available data;
- Frequency of reporting, integration with other surveillance systems;
- Existence, frequency and quality of monitoring.

Surveillance of other diseases

- Maternal and neonatal mortality.

Vaccine safety surveillance systems

- Existence of training, surveillance of vaccination of pregnant women on AEFI, available data.

Other vaccines than tetanus in pregnancy

- Policy, partners, and delivery mechanisms for vaccines to pregnant women other than tetanus, e.g. influenza, pertussis, yellow fever, and meningococcus A, available data;
- Main barriers for introducing additional vaccines for pregnant women, by administration level;
- Potential interventions to support uptake of maternal vaccinations, e.g. elimination of user fees, client/provider communication, availability of medicines.

**Figure 4** Study phase 3: variables collected from interviews of 30 selected LMICs. A semistructured questionnaire will be used to assess the preparedness of antenatal care services for introducing (additional) immunisations for pregnant women in selected low-income and lower-middle-income countries, and to understand the strengths and weaknesses of current immunisation to guide future planning. Internal validation questions are incorporated and probing for further details will be done when deemed necessary by the interviewer(s). Sources of data provided are requested, that is, if administrative data or personal estimates. AEFI, adverse events following immuni; ANC, antenatal care; EPI, Expanded Programme on Immunization; HMIS, health management information system; LMICs, low-income and middle-income countries; NITAG, national immunisation technical advisory group; Td, tetanus diphtheria; TT, tetanus toxoid.

**Supply side (health system)**

Service delivery

- Integration of antenatal care (ANC) and Expanded Programme on Immunization (EPI);
- Accessibility, outreach services;
- Costs of services;
- Availability, supply chain;
- Quality and mode of delivery;
- Cultural appropriateness;
- Follow-up, e.g. mobile technology;
- Function of referral system.

Health care workers

- Education, professional skills;
- Workload, working conditions;
- Professional attitudes (non-discriminatory);
- Communication skills;
- Role of community health workers.

Information

- Actionable health information system;
- Demand side information campaigns.

Medical products, vaccines, technology

- Safety;
- Supply chain skills, documentation.

Financing

- Domestic, external funding;
- Devolution of health services planning and financing;
- Results-based approaches.

Leadership, governance

- Partnerships;
- Political priorities;
- Health system organisation, e.g. level of decentralisation;
- Accountability mechanisms;
- Community participation.

**Demand side (pregnant women)**

Socio-cultural and -economic factors

- Socio-economic status;
- General health literacy;
- Knowledge about maternal immunization;
- Mobility, security;
- Personal characteristics, i.e. age, marital status, parity;
- Culture, religion.

Health systems interaction

- Reception of adequate information;
- Distance to health facility;
- Direct and indirect costs of services;
- Transport, infrastructure (safety, accessibility);
- Opportunity costs, i.e. time spent at facility;
- Clarity of procedures;
- Communication (trust);
- Non-discrimination;
- Community outreach.

**Figure 5** Study phase 4: country study analysis framework for 10 country visits. Key informant interviews, health facility visits and focus group discussions will enable observation and collection of further data on the variables from the previous study phases, in particular at different levels of the healthcare system, and of sociocultural and socioeconomic factors. End-users, that is, pregnant women, will not be interviewed as it would require a separate study design, and their perspective will be indirectly included through the participation of community health workers at stakeholder meetings.

protected at birth against neonatal tetanus, with a cut-off of 90%, and high and low ANC performance (with a cut-off of the median ANC4+ coverage in countries with available data). PAB was identified as a more reliable proxy measure than TT2+, as the issue of not including already vaccinated women in the numerator used for estimating the latter indicator would be avoided. The PAB cut-off level was set based on the target required to attain and sustain MNTE, whereby >80% of pregnant women are immunised against tetanus. The country selection will include a representation of all MI delivery models and WHO regions, with a focus on Africa and South-East Asia where maternal and neonatal mortality are highest, and will ensure inclusion of high-performing countries in order to include likely early adopters of new maternal vaccines and learning cases of best practices.

The interviews will collect data on the policy, governance and funding environment for EPI and ANC programmes, ANC delivery and maternal tetanus vaccination including monitoring and evaluation of results. The questionnaire will be shared with WHO Country Office focal points and MoH MNCH and EPI managers for compilation in advance of the teleconference, allowing for discussion and clarification when needed during the

actual interview. Responses will be recorded using standard data entry procedures, and may be voice-recorded if consent is obtained by the interviewees. Any discordant responses will be attempted to be resolved by consensus, and incomplete responses will be followed-up. A summary of key findings will be shared with the participants to confirm the responses were correctly captured.

### Country visits (data collection phase 4)

Finally, in-country visits will be conducted in order to collect data on MI from key decision-makers and implementers at every level of the healthcare system, as well as to determine actual delivery, capacity and coordination of ANC and EPI services, on both supply and demand sides of the healthcare services (figure 5). Ten countries will be selected based on the high, medium or low performance of MI systems as assessed by PAB and TT2+, a range of different MI delivery models (eg, degree of coordination between EPI and ANC in MI delivery), and agreement by senior national and subnational MNCH and EPI staff for study visits. The final country selection will ensure representation of the range of MI delivery models, and will include high-performing countries, MNTE priority countries and countries with high ANC4+ coverage. Site visits will include ANC and EPI sites and session observations, focus group discussions and in-depth interviews. The week-long visits will be piloted in two countries to adjust and refine the data collection tools and the standard operating procedures, and data from these two countries will be included in the final analysis.

An initial joint focus group discussion will be held with national-level stakeholders, followed by key informant interviews with stakeholders pertinent to MI, ANC and EPI services at subnational levels of the healthcare system, including decision-makers and policy-makers, technical and financial parties, and civil society, such as non-governmental organisations. The study will aim to conduct a total of 12 health facility visits taking into account a balance of geographical locations, urban and rural areas, and—if possible—different types of health facilities (eg, small and large health units). The country visits will be concluded with an on-site debriefing and joint data analysis with MoH MNCH and EPI focal points and other main country-level stakeholders. End-users, that is, pregnant women, will not be interviewed as it would require a separate study design; however, their perspective will be indirectly included through the participation of community health workers at stakeholder meetings.

### Data analysis plan

The cross-sectional data analyses will be carried out over four data collection phases (desk review of global data, online questionnaire, in-depth country interviews and country visits). The first three will yield quantitative data. The last two data collection phases will also provide an in-depth qualitative analysis of data collected from a select number of countries. Below, we describe the analyses for each phase.

### Desk review of global data (phase 1)

The MIACSA project will conduct a desk review of global databases (JRF, United Nations mortality reports, DHS, MICS, WHO MNCAH policy survey database, MNTE database, WHO/Unicef estimates of national immunization coverage) targeting 136 LMICs.

The primary outcome variable (dependent variable) to asses MI performance will be PAB (cut-off level <90% and ≥90%) and the independent variables will include country economic level, immunisation coverage, mortality, service coverage, available ANC and vaccination policies and availability of a National immunisation Advisory Committee (figure 2).

We will first asses the database for completeness of data. We will also conduct a sensitivity analysis based on imputation of data based on available predictors for countries with missing data on PAB. Results from the complete case analysis will be compared with the sensitivity analysis to explore bias due to missing data.

We will conduct bivariate analyses to assess whether the dependent variables are associated with the independent variables. We will also do multivariable analyses within subgroups, since vaccinations may differ by other factors (eg, WHO Regions; Global Alliance for Vaccines and Immunization (GAVI) status; World Bank income level; MNTE and female literacy rate).

For continuous variables, we will first assess the normality using the Shapiro-Wilk test. If needed, we will make appropriate transformations to normalise the data or group them into categories as necessary. We will then compare the distributions of the variables by groups with two-sided $\chi^2$ (categorical variables) or t-tests (continuous variables) or the equivalent non-parametric tests (eg, Fisher's exact or Wilcoxon/Kruskal-Wallis), as appropriate. A two-sided p value of 0.05 will be considered significant.

To create a multivariable model, we will include all variables that are significantly associated with the dependent variable and those variables that have shown association within the available literature. We will then asses for collinearity and remove one of the variables if collinearity is found. We will also assess for interactions and will create interaction terms to be included in the model if any interactions are found. Both forward and backward elimination will be conducted to assess the goodness of fit and create the final model.

### Global online survey (phase 2)

The variables are based on the online survey as described in the Desk review of global data (phase 1) section. For a summary of the included components see figure 3. Data from the online survey will be checked for completeness and consistency and coded to reflect skip patterns. The complete data set for analyses will include PAB from phase 1 (desk review database) and will be linked with the database containing responses for the global online questionnaire. Descriptive analyses will be conducted including summary measures. Bivariate

analyses will be conducted to assess the associations between the questionnaire variables and the dependent variable PAB and the significance of the relationship will be tested with Fisher's exact test. Logistic regression models will be used to assess the relationship between the responses in the online questionnaire and high coverage of PAB≥90% independently. These models will be created as described in phase 1. A two-sided p value of 0.05 will be considered significant.

### Telephone conferences (quantitative analyses, phase 3)

The primary objective is to provide descriptive information about MI services and its organisation (figure 4).

Data from this phase will be checked for completeness and consistency and coded to reflect skip patterns. Descriptive analyses will be conducted including summary measures.

### Qualitative analyses based on country visits and telephone conferences (phase 4)

Ten countries will be visited to conduct qualitative interviews at national and subnational level. Resulting qualitative information from these visits as well as from telephone interview conducted in the previous phase will be used in a thematic analysis applied by trained qualitative data analysts to the following qualitative data sources: comments and free-text responses to telephone interview questions (phase 3), semistructured interviews with community health workers (phase 4), comments and free-text response to stakeholder and facility manager interviews (phase 4) and comments provided during debrief sessions with national level stakeholders in-country (phase 4). Thematic analysis will be applied to intracase and cross-case analysis. First, an intracase analysis will organise and reduce qualitative findings within each country along two criteria: (1) relevance of finding to research questions and (2) relative frequency of finding across data sources. Second, a cross-case analysis will organise findings across countries into themes generated from research questions and subthemes generated from grounded analysis of data collected. Two qualitative data analysts will co-organise and reduce intracountry findings. For cross-country findings, qualitative analysts will independently generate themes and subthemes for cross-case analysis and will then resolve any intercoder divergence in themes and subthemes based on the relevance of theme to data source, relevance of theme to research questions and robustness of theme relative to alternative themes. See figures 4 and 5 for the included components.

### Consolidated data analysis

To inform the development of a typology of MI delivery models approaches in LMICs quantitative and qualitative data analysis results will be consolidated in a global analysis of MI and ANC service delivery approaches in countries as well as individual country profiles that shall support countries to conduct self-assessments of their MI and ANC systems strengths and capacity gaps. On the basis of the advice of the project's advisory group, a checklist approach will be considered to provide a profiling for countries with sufficient data available, including indicators on policy and governance, financing, programme management, service delivery systems and demand-side issues. Ultimately, such a profiling shall help countries and other MI stakeholders to identify the needs for targeted support to strengthen existing MI programmes or to reach readiness to introduce future programmes.

### Limitations

The data analyses will take into account the limitations of the study, including the reliability of the selected outcome measures, that is, PAB, potential biases introduced by the limited number of countries for which in-depth information will be available, that is, through telephone interviews and in-country visits, selective sampling of in-country site visit locations, missing data and the fact that the end-users' perspective will be captured only indirectly through facility-based and community-based health workers.

## ETHICS AND DISSEMINATION
### Ethical considerations

The first three phases of the study are exempt from ethical permission as participants will provide information on operations and administration of public health services on a purely professional basis, and without disclosure of person-identifiable information.

Country ownership will be ensured by transferring the responsibility for providing data to in-country focal points, and by joint, on-site analysis of the data collected during the country visits with the main stakeholders. The study aims to contribute to the evidence needed to ensure more equitable access to high-impact global health interventions, such as MI.[25]

### Data management and dissemination

The data will be managed and analysed by data clerks who were not part of the data collection. Anonymised data from surveys and key informant interviews, excluding any confidential information as identified by the in-country focal points, will be uploaded to a publicly available data repository hosted by the WHO. Recordings from country interviews will be transcribed before the qualitative analyses and destroyed at the completion of the data analyses.

The results will be submitted for publication in peer-reviewed journals, as well as in a MIACSA project report that will be shared widely with global health decision-makers, researchers, product developers and implementers. The report and/or specific aspects of the project will be presented at international stakeholder meetings, with the ultimate aim to establish a knowledge network of countries exploring MI implementation strategies. Further, the

results will be shared through summaries on the WHO website and in public fora.

To ensure wide distribution of the project findings to the international scientific community and national stakeholders involved in MI, findings will be also shared at the end of the project through a large stakeholder convening. At this meeting, key aspects of maternal tetanus vaccination service delivery mechanisms and ANC capacities identified in select countries will be discussed to enable exchange of lessons learnt between select participating countries and to discuss generalisable lessons learnt that may improve MI service delivery through an integrated platform considering immunisation and maternal child healthcare mechanisms.

Dissemination of the MIACSA results will aim to provide advice on best practices, policy requirements, capacity needs and health system changes needed for successful introduction and integration of new maternal vaccines into national health systems, including ANC and EPI services, in LMICs.

**Author affiliations**
[1]Department of Maternal Newborn Child and Adolescent Health (MCA), Epidemiology Monitoring and Evaluation (EME), World Health Organization, Geneva, Switzerland
[2]Department of Immunization, Vaccines and Biologicals (IVB), Initiative for Vaccine Research (IVR), World Health Organization, Geneva, Switzerland
[3]Independant consulting and advisory group, MMGH Consulting GmbH, Zürich, Switzerland
[4]Department of Infectious Disease Epidemiology, Robert Koch Institute, Berlin, Germany
[5]Faculty of epidemiology and population health, Department of infectious disease epidemiology, London School of Hygiene and Tropical Medicine, London, UK
[6]Department of Pediatrics, Section of Infectious Diseases, Baylor College of Medicine, Houston, Texas, USA
[7]Department of Obstetrics and Gynaecology, Monash University and Monash Health, Melbourne, Australia

**Acknowledgements** The investigators wish to thank the EPI and MNCH focal points at the regional and national WHO offices for their helpful contributions to the planning of the study, and the external specialists of the Expert Advisory Panel for their valuable advice and guidance on the development of the protocol methodology. The authors also wish to thank Dr Peter Mark Jourdan for assistance in writing and editing the manuscript.

**Contributor** The MIACSA expert advisory panel (EAP) consists of the following members: Flor M Muñoz (chair), Michelle L Giles (co-chair), Mercy Ahun, Martina Baye and Matthews Mathai. Observers to the EAP meetings included: Carsten Mantel, Elizabeth Mason, Sonja Mertens, Jayani Pathirana and Sara Rendell. Additional WHO experts included Emily Wootton, Laura Nic Lochainn and Ahmadu Yakubu.

**Contributors** NR and PL designed the study with inputs from TD and JH; CM, EM, FMM, MG, TD, JH, ACM and the MIACSA drafted the protocol with NR and PL; and all the authors reviewed and approved the final manuscript version.

**Funding** This work was supported by the Bill & Melinda Gates Foundation, grant number OPP1156378.

**Competing interests** None declared.

**Patient consent for publication** Not required.

**Ethics approval** The protocol for the country visits was approved by the WHO Research Ethics Review Committee (ERC.0002908).

**Provenance and peer review** Not commissioned; externally peer reviewed.

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
