## [Reviewer comments · BMJ Open]

ARTICLE DETAILS

TITLE (PROVISIONAL)	The Maternal Immunization and Antenatal Care Situation Analysis (MIACSA) study protocol: A multi-regional, cross-sectional analysis of maternal immunization delivery strategies to reduce maternal and neonatal morbidity and mortality
AUTHORS	Roos, Nathalie; Lambach, Philipp; Mantel, Carsten; Mason, Elizabeth; Munoz, Flor M.; Giles, Michelle; MIACSA, expert advisory panel group; Moran, A; Hombach, Joachim; Diaz, Theresa

VERSION 1 - REVIEW

REVIEWER	Joses Jain Columbia University Medical Center, United States
REVIEW RETURNED	11-Jul-2018

GENERAL COMMENTS	The authors present an ambitious study protocol to evaluate factors related to maternal vaccine delivery in low and middle-income countries, using delivery of the tetanus vaccine as an example. Overall, this protocol is well-written and thoughtful consideration has been given to potential confounders and limitations. The only suggestion is that the statistical analysis plan is somewhat vague as written and may benefit from some more specifics pertaining to each data set that will be collected. Ultimately, the results of this study could prove very important in developing strategies to improve global maternal health.
---

REVIEWER	Paula Broeiro-Gonçalves Family Physician - UCSP Olivais Lisbon, Medicine Faculty invited teacher - Lisbon; Clinical expert of Portuguese Medicines Agency - INFARMED, Lisbon Epidemiology PhD student - National Public Health School, Lisbon.
REVIEW RETURNED	24-Jul-2018

GENERAL COMMENTS	•The title suggests a longitudinal study – It should be reviewed•Abstract: Given the prospect of introducing new maternal vaccines in the near future, it is essential to identify and understand current policies, practices, and unmet needs for introducing and/or scaling up MI in low and middle-income countries (LMICs).
--

	–This is not in accordance with the aim - to determine how existing health care services can be further strengthened to improve maternal and neonatal outcomes, and how they could accommodate new MI vaccines. •Introduction: – Before to respond to the objective some questions should be answered:  1. What do health services ensure good maternal immunization? 2. What strategies are used? Are these measures generalizable? 3. What health services cannot ensure good pre-natal maternal immunization? Why (include social health determinants)? – The authors should reflect on the ethical issues related with:  1. The creation of new need without assuring that basic vaccines were right implemented. 2. The prioritisation of different maternal immunization by country – The objective how they could accommodate new MI vaccines turns the study more complex. I suggest this aim for other studies after countries immunisation prioritisation (e.g., Group B Streptococcus in Nigeria). • Data analyses: – This section need review after the clarification the above points Conclusion: Despite the objections this study will have an unquestionable social value and for public health. We suggest to authors:  1. A critical review 2. The clarification of the doubts 3. To argue the points of disagreement After a major reflective review, this paper could be published.
--	--

VERSION 1 – AUTHOR RESPONSE

Reviewer responses

Reviewer 1: Joses Jain	
Institution and Country: Columbia University Medical Center, United States	
Comment	Response
The authors present an ambitious study protocol to evaluate factors related to maternal vaccine delivery in low and middle-income countries, using delivery of the tetanus vaccine as an example. Overall, this protocol is well-written and thoughtful consideration has been given to potential confounders and limitations. The only suggestion is that the statistical analysis plan is somewhat vague as written and may benefit from some more specifics pertaining to each data set that will be collected. Ultimately, the results of this study could prove very important in developing strategies to improve global maternal health.	We thank the reviewer for the suggestion. We have expanded the section on data analysis plan in the manuscript. See lines 253 – 372 in the manuscript version with track changes.

Reviewer 2 : Paula Broeiro-Gonçalves

Institution and Country: Family Physician - UCSP Olivais Lisbon, Medicine Faculty invited teacher - Lisbon; Clinical expert of Portuguese Medicines Agency - INFARMED, Lisbon Epidemiology PhD student - National Public Health School, Lisbon.

Comments

Responses

The title suggests a longitudinal study – It should be reviewed

This study is not longitudinal as we are not following countries and examining change over time. The title in its current form clarifies that the study is cross sectional in design as for each phase of the project data collection occurs only once, for example during completion of an online survey (phase 2) or during telephone interviews (phase 3).

Abstract: Given the prospect of introducing new maternal vaccines in the near future, it is essential to identify and understand current policies, practices, and unmet needs for introducing and/or scaling up MI in low and middle-income countries (LMICs).
–This is not in accordance with the aim - to determine how existing health care services can be further strengthened to improve maternal and neonatal outcomes, and how they could accommodate new MI vaccines.

Thank you for this comment. We have adjusted the aims of the project described in the Introduction section to be harmonized with what is written in the abstract. See lines 131-134 in the version with track changes.

Introduction: Before to respond to the objective some questions should be answered:

1. What do health services ensure good maternal immunization?
2. What strategies are used? Are these measures generalizable? 3. What health services cannot ensure good pre-natal maternal immunization? Why (include social health determinants)?

Thank you for the questions to help frame the objective of the project. I will respond to the three questions point-by point.

1. Given the need for collaboration between two programs (EPI and ANC) to ensure pregnant women are vaccinated, there is a need to better understand what the determinants are for high coverage of maternal immunization. The MIACSA project aims to understand this from the global perspective through a desk review of globally available data and conducting an online survey in Low and Middle Income Countries.

2. MI strategies are not well defined as there are no globally recognized definitions for MI delivery strategies. The MIACSA project aims to further clarify, and if possible, categorize countries according to MI delivery strategies and their specific characteristics in relation to efficacy (PAB coverage levels).

3. This is an important question. An aim of the MIACSA project is indeed to try and

	understand what factors may contribute to high or low coverage for maternal immunization and the social determinants which may contribute to this. Social determinants at the individual level will not be collected as the project focuses on national level data primarily, and during the country level visits on a small convenience sample of selected health facilities.
The authors should reflect on the ethical issues related with:  1. The creation of new need without assuring that basic vaccines were right implemented. 2. The prioritization of different maternal immunization by country – The objective how they could accommodate new MI vaccines turns the study more complex. I suggest this aim for other studies after countries immunization prioritization (e.g., Group B Streptococcus in Nigeria). 	Thank you for bringing up the ethical issues of introducing a new vaccine in the country. And we also welcome the suggestion of looking deeper into prioritization of maternal vaccines. The MIACSA project does not aim to introduce any new vaccines to the visited countries, but to conduct an assessment of the status quo of service delivery of Maternal Neonatal Tetanus vaccination. Before the introduction of a new vaccine into a country, there are many important considerations as outlined by previous WHO
	publications.¹. There are additional considerations when contemplating maternal vaccines such as existing antenatal care services quality and capacity. Another goal of the MIACSA project is to have a better understanding of the existing gaps and needs within existing antenatal care services. . 2. Although prioritization of new vaccines is very important for implementation decision making, the MIACSA project does not aim to make recommendations to countries in regard to this.
Data analyses: – This section need review after the clarification the above points	The data analysis plan section has been reviewed to be more detailed, however the above comments does not impact the data analysis plan. The above issues are rather to be discussed in the project report. See lines 253 –

	372 for the updated data analysis plan in the version with track-changes.
Conclusion: Despite the objections this study will have an unquestionable social value and for public health. We suggest to authors:  1. A critical review 2. The clarification of the doubts 3. To argue the points of disagreement After a major reflective review, this paper could be published.	We thank the reviewer for all the pertinent questions which we have addressed in a point-by point manner and also made the necessary changes to the manuscript.

VERSION 2 – REVIEW

REVIEWER	Joses Jain United States
REVIEW RETURNED	21-Jan-2019

GENERAL COMMENTS	This is an important study protocol that aims to provide valuable information regarding current practices and areas for improvement in morbidity and mortality with respect to preventable illness. The study aims to collect information from multiple modalities to provide current information regarding practices of maternal vaccine administration in low and middle income countries. This updated version of the manuscript provides further detail regarding anticipated statistical analyses of the collected data, which will require additional review upon completion.
---

REVIEWER	Paula Broeiro-Gonçalves Medicine Faculty, Lisbon University Health Ministry, UCSP Olivais, ACES Lisboa Central, ARSLVT
REVIEW RETURNED	21-Oct-2018

GENERAL COMMENTS	Thank you for ameliorating the manuscript
---

VERSION 2 – AUTHOR RESPONSE

2. Reviewer: 2 (Paula Broeiro-Gonçalves)

- Thank you for ameliorating the manuscript

Response: No action required

3. Reviewer: 1 (Joses Jain)

- This is an important study protocol that aims to provide valuable information regarding current practices and areas for improvement in morbidity and mortality with respect to preventable illness. The study aims to collect information from multiple modalities to provide current information regarding practices of maternal vaccine administration in low and middle income countries. This updated version of the manuscript provides further detail regarding anticipated statistical analyses of the collected data, which will require additional review upon completion.

Response: No action required